# The droplet formation-dissolution transition in different ensembles: Finite-size scaling from two perspectives

**Franz Paul Spitzner[1][*], Johannes Zierenberg[1,2,3] and Wolfhard Janke[1]**

**1** Institut für Theoretische Physik, Universität Leipzig,
Postfach 100 920, 04009 Leipzig, Germany
**2** Max Planck Institute for Dynamics and Self-Organization,
Am Fassberg 17, 37077 Göttingen, Germany
**3** Bernstein Center for Computational Neuroscience,
Am Fassberg 17, 37077 Göttingen, Germany

[*] spitzner@itp.uni-leipzig.de

## Abstract

The formation and dissolution of a droplet is an important mechanism related to various nucleation phenomena. Here, we address the droplet formation-dissolution transition in a two-dimensional Lennard-Jones gas to demonstrate a consistent finite-size scaling approach from two perspectives using orthogonal control parameters. For the canonical ensemble, this means that we fix the temperature while varying the density and vice versa. Using specialised parallel multicanonical methods for both cases, we confirm analytical predictions at fixed temperature (rigorously only proven for lattice systems) and corresponding scaling predictions from expansions at fixed density. Importantly, our methodological approach provides us with reference quantities from the grand canonical ensemble that enter the analytical predictions. Our orthogonal finite-size scaling setup can be exploited for theoretical and experimental investigations of general nucleation phenomena – if one identifies the corresponding reference ensemble and adapts the theory accordingly. In this case, our numerical approach can be readily translated to the corresponding ensembles and thereby proves very useful for numerical studies of equilibrium cluster formation, in general.



## Contents


# 1 Introduction

There is a wide range of nucleation processes in nature, with the formation of a droplet being the prime example [1–3]. In principle, the formation of a droplet in an equilibrium vapour can be induced by a pressure or temperature quench, corresponding to a supersaturated or supercooled gas. Due to experimental constraints on controlling the temperature homogeneously, most experimental approaches follow a setup at fixed temperature with increasing oversaturation [4], whereas undercooling is typically used for crystal nucleation in metals and colloids [5,6].

On the quest to develop a theory on droplet formation, intensive research was devoted to the phenomenological understanding of droplet formation and growth [1–3,7]. The resulting classical nucleation theory is still extensively employed, where droplet growth uses particles from a reservoir. Then, the stability of a droplet is determined by the competition between energy gain from the droplet bulk and surface tension from the resulting interface. This theory is particularly successful for the scenario of small metastable droplets, which one would expect to observe on the time scale of an experiment due to their low free-energy barrier. Recently, molecular dynamics simulations with $O(10^9)$ Lennard-Jones atoms have shown homogeneous nucleation of small metastable droplets with results comparable to experiments [8].

It was relatively late that the problem was formulated in an equilibrium framework including the droplet's surrounding [9–12]. Here, the energy gain of forming the droplet competes with the cost of forming the interface, as well as the entropic loss by binding otherwise free gas particles to the droplet. As a result, one expects either a gas phase or a mixed phase with a single macroscopic droplet in equilibrium with the surrounding vapour. The free-energy barrier separating both phases increases with system size [10–12] and the probability to observe increasingly large droplets during a given time decreases drastically. In this work, we focus on this equilibrium scenario.

Monte Carlo simulations are an established tool to study the equilibrium scenario of droplet formation either at fixed temperature (varying density) [11,13–21] or at fixed density (varying temperature) [22–24]. This requires techniques that overcome the large free-energy barrier, or systems need to be prepared suitably. So far, the different approaches yielded overall consistent results of the leading-order finite-size scaling corrections. Yet, different formalisms and models made direct comparisons of the two perspectives cumbersome. Here, we aim to close this gap

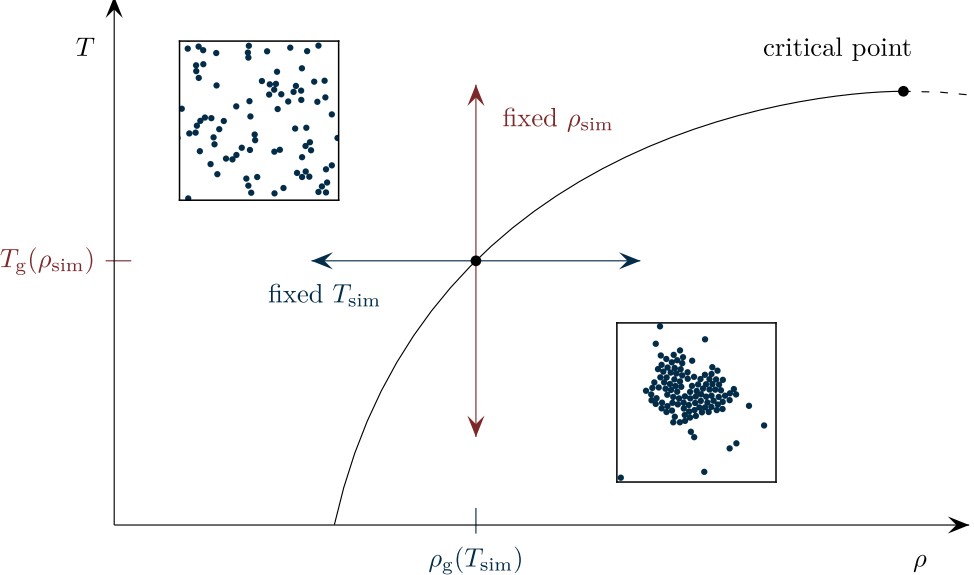

Figure 1: Sketch of the transition between the pure gas phase and the mixed phase of a liquid droplet surrounded by vapour. Below the critical point, the black infinite-size transition line can be crossed in either one of two *orthogonal regimes*: The blue horizontal arrow depicts the fixed-temperature approach in which density serves as the control parameter, where $\rho_g(T_{\text{sim}})$ is the infinite-size transition density. Alternatively, the red vertical arrow depicts the fixed-density approach in which temperature serves as the control parameter and $T_g(\rho_{\text{sim}})$ is the infinite-size transition temperature.

by taking one model subject to both temperature and density variation. We choose the two-dimensional Lennard-Jones gas to numerically verify the phenomenological theory by Biskup et al. [10] at fixed temperature, which was rigorously proven only for the two-dimensional lattice gas.

The remainder of the paper is structured as follows: We recapitulate the theory on droplet formation by Biskup et al. [10] and its extension [23] in the next section. The model, our methods and implementation details are discussed in Sec. 3, followed by the results in Sec. 4. Lastly, Sec. 5 contains a concluding discussion.

## 2 Theory

Let us consider a particle gas in a canonical ensemble of fixed particle number ($N$), fixed volume ($V$) and fixed temperature ($T$). For such a system, we could induce droplet formation by choosing any of the three as a control parameter. As an example, when taking the gas to be in a fixed volume, we could either increase the particle number or decrease temperature to form a droplet (see Fig. 1).

In particular, fixing $V$ and $T$ while considering a variable density $\rho = N/V$ allows us to construct a reference grand canonical ensemble in which we can define the bulk (or background) densities of a system in a gas or liquid phase. Thus, at coexistence,

$$\rho_g = \frac{N_g}{V} \qquad \text{and} \qquad \rho_l = \frac{N_l}{V} \tag{1}$$

are the expected densities for the system in the respective pure phases (Fig. 2). We then choose $N$ as the control parameter in the canonical ensemble; but only in light of the reference grand

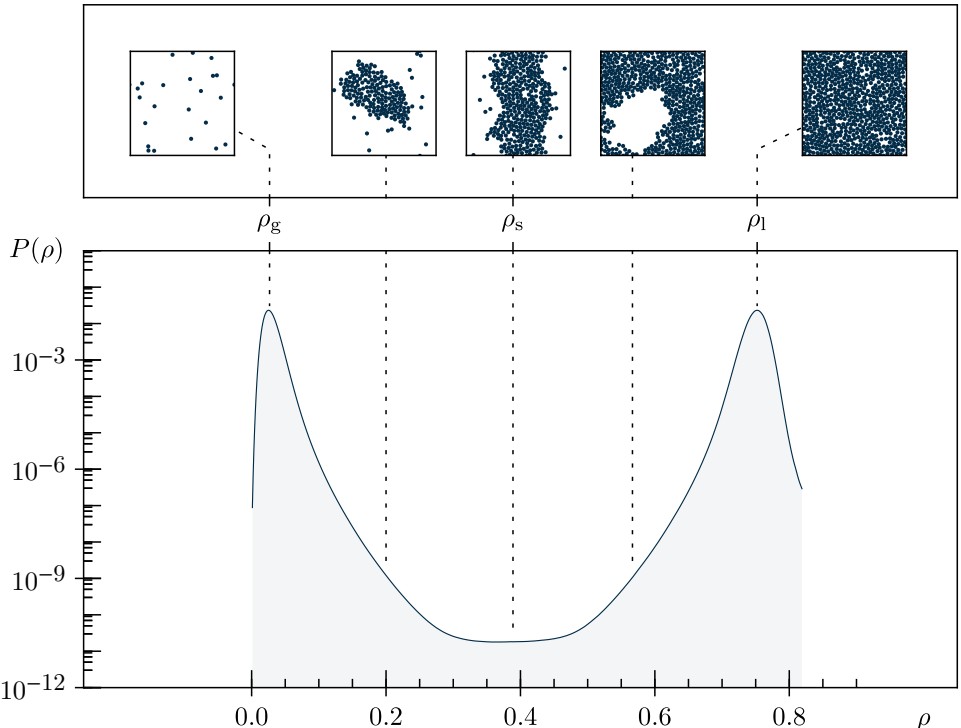

Figure 2: Grand canonical probability distribution of a system with linear size $L = 30$ at temperature $T = 0.4$ and equal-height chemical potential $\mu \approx -1.6518$. Matching snapshots are shown on top and the whole density region over which the gas-liquid transition takes place is covered. $P(\rho)$ shows peaks at the densities corresponding to the gas ($\rho_g$) and the liquid ($\rho_l$). In between appears a suppressed plateau around the density where a stripe occurs ($\rho_s$).

canonical ensemble can we define a *particle excess* over the background gas. This particle excess ($\delta N$) either manifests itself through local density fluctuations of the gas ($\delta N_F$) or through the formation of a macroscopic droplet ($\delta N_D$):

$$\delta N = N - N_g = \delta N_D + \delta N_F. \tag{2}$$

Biskup et al. have shown that "the probability of even a single droplet of the intermediate scale is utterly negligible" [10], i.e., that the droplet excess $\delta N_D$ will only contribute to the creation of a single, large droplet – as opposed to a multitude of small or intermediately sized ones. Consequently, the whole discussion of the droplet formation-dissolution transition is simplified significantly and can be expressed by a two-state model.

When further utilising the bulk densities, the amount of excess within the droplet can be related to its volume $V_D$ through

$$\delta N_D = \left( \rho_l - \rho_g \right) V_D. \tag{3}$$

Using the particle number within the droplet ($N_D = \rho_l V_D$), we can also introduce the *droplet fraction*

$$\lambda = \frac{\delta N_D}{\delta N} = \frac{\left( \rho_l - \rho_g \right)}{\left( \rho - \rho_g \right) \rho_l} \frac{N_D}{V} = \lambda(\rho, V, N_D), \tag{4}$$

serving as an order parameter. This quantity $\lambda$ is most intuitively thought of as a rescaled measure of droplet size, relative to the present excess $\delta N$ ($\lambda \in [0, 1]$).

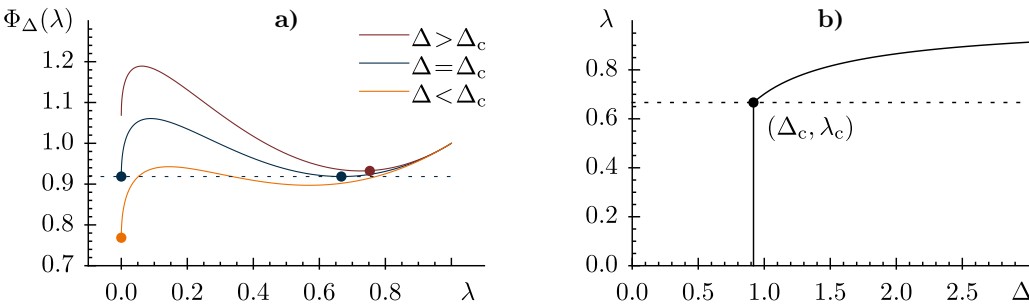

Figure 3: **a)** Free-energy functional $\Phi_\Delta(\lambda)$ for different rescaled densities $\Delta$. At the transition density, phase coexistence is indicated through the two solutions, the oversaturated vapour with no droplet ($\lambda = 0$) and the equilibrium droplet with surrounding vapour ($\lambda_c = 2/3$). **b)** Plot of the analytic solution for the rescaled droplet size $\lambda(\Delta)$.

Compellingly, the theory suffices with considering free-energy contributions that arise from the single droplet ($F_D$) and from density fluctuations ($F_F$) in the surrounding gas phase. They are approximated by $F_D = \tau \sqrt{V_D}$ and $F_F = (\delta N_F)^2 / 2\hat{\kappa} V$ [10], where we have assumed for $F_D$ the special case of two dimensions. The constants $\tau$ and $\hat{\kappa}$ stem from the reference ensemble and quantify the surface free energy per unit volume of the droplet and the (reduced) isothermal compressibility. The equilibrium solution is obtained by minimising the total free energy of the system

$$F = F_D + F_F = \tau \sqrt{\frac{\delta N}{\rho_l - \rho_g}} \left( \sqrt{\lambda} + \Delta (1-\lambda)^2 \right) = \tau \sqrt{\frac{\delta N}{\rho_l - \rho_g}} \Phi_\Delta(\lambda), \tag{5}$$

where

$$\Delta = \frac{\sqrt{\rho_l - \rho_g}}{2\hat{\kappa}\tau} \frac{(\delta N)^{3/2}}{V} = \frac{\sqrt{\rho_l - \rho_g}}{2\hat{\kappa}\tau} \left(\rho - \rho_g\right)^{3/2} \sqrt{V} = \Delta(\rho, V) \tag{6}$$

is the *rescaled density parameter*. In simple words, $\Delta$ describes how much the current density is increased over the background gas density. Comparing with Fig. 3 a), one readily sees that as long as the excess is below the threshold ($\Delta < \Delta_c$), the functional $\Phi_\Delta(\lambda)$ is minimal for a vanishing droplet fraction – which corresponds to the oversaturated vapour with no droplet ($\lambda = 0$). Reaching the transition at $\Delta_c$, a droplet of leading-order size $\lambda_c$ is formed:

$$\Delta_c = \frac{3}{4}\sqrt{\frac{3}{2}} \approx 0.918 \qquad \text{and} \qquad \lambda_c = \frac{2}{3}. \tag{7}$$

For higher densities ($\Delta > \Delta_c$), the analytic solution [14] is

$$\lambda = \frac{4}{3} \cos^2 \left( \frac{\pi - \arccos\left(\frac{3\sqrt{3}}{8\Delta}\right)}{3} \right). \tag{8}$$

Note that the actual numeric value of the transition point is only determined by dimension ($\Delta_c = \frac{1}{d}\left(\frac{d+1}{d}\right)^{(d+1)/d}$, $\lambda_c = \frac{2}{d+1}$) [10]. The leading-order finite-size corrections are encapsulated in the definition of $\lambda$ and $\Delta$. While the shape of $\lambda(\Delta)$ in Eq. (8) and Fig. 3 b) again depends on dimension, the qualitative interpretation is universal: For rescaled densities $\Delta < \Delta_c$, the fraction of excess in the droplet is zero ($\lambda = 0$) – corresponding to zero droplet size $N_D = 0$.

For $\Delta > \Delta_c$ the fraction of excess in the droplet has a non-zero value ($\lambda > \lambda_c$) – corresponding to a macroscopic droplet.

By plugging in the finite-size transition density $\rho = \rho_c(V)$ into Eq. (6) with $\Delta(\rho_c(V), V) = \Delta_c$, the leading-order finite-size scaling of the native transition density, cf. [11, 12], follows as

$$\rho_c(V) = \rho_g + a_\rho V^{-1/3} \qquad \text{with} \qquad a_\rho = \left( \frac{2\hat{\kappa}\tau\Delta_c}{\sqrt{\rho_1 - \rho_g}} \right)^{2/3}. \qquad (9)$$

We can also use the density scaling to find out how the droplet scales at the transition point. To that end, we rewrite the definition of $\lambda$ [Eq. (4)] such that

$$N_D(\rho, V) = \lambda \left( \rho - \rho_g \right) \left( \frac{\rho_1}{\rho_1 - \rho_g} \right) V. \qquad (10)$$

Directly at the transition, we know that $\lambda = \lambda_c$ and $\rho = \rho_c(V)$. The scaling behaviour of the number of particles in the largest droplet – at the transition point – can then be obtained by replacing $(\rho_c(V) - \rho_g)$ using Eq. (9):

$$N_D(\rho_c(V), V) = a_{N_D} V^{2/3} \qquad \text{with} \qquad a_{N_D} = a_\rho \lambda_c \left( \frac{\rho_1}{\rho_1 - \rho_g} \right). \qquad (11)$$

Up to this point, the bulk densities $\rho_g$, $\rho_1$ as well as $\hat{\kappa}$ and $\tau$ were assumed to be constant, which is only valid for large systems – and fixed temperatures; the transition was driven by density. However, it was shown that the leading-order finite-size behaviour is identical for the temperature-driven transition [23]. Once the temperature dependence of the quantities is restored in Eq. (6), one can rewrite it as

$$\Delta^{2/3} V^{-1/3} = \left( \rho - \rho_g(T) \right) \left( \frac{\sqrt{\rho_1(T) - \rho_g(T)}}{2\hat{\kappa}(T)\tau(T)} \right)^{2/3} = f(\rho, T). \qquad (12)$$

When expanding this function $f(\rho, T)$ around the infinite-size transition temperature $\lim_{V \to \infty} T_c(V) = T_g$, we see that the first term vanishes in $f(\rho, T) = f(\rho, T_g) + f'(\rho, T_g)(T - T_g) + ...$, since $\rho_g(T_g) = \rho$. We can then solve for the finite-size transition temperature $T = T_c(V)$ and obtain the finite-size scaling of the transition temperature at fixed density:

$$T_c(V) \simeq T_g + a_T V^{-1/3} \qquad \text{with} \qquad a_T = \frac{\Delta_c^{2/3}}{f'(\rho, T_g)}, \qquad (13)$$

where $f'$ denotes the temperature derivative $\partial f / \partial T$. Generally speaking, $a_T$ remains unsolved because the actual temperature dependence of individual quantities is unknown. However, $f'(\rho, T_g)$ can be approximated numerically and, as we will show in Sec. 4.1, finite-size corrections to the contributing reference quantities (and thus $f'$) are negligible.

## 3  Model and Methods

### 3.1  Lennard-Jones Gas in Two Dimensions

The Lennard-Jones gas is constructed using point particles that can move freely in a domain of linear size $L$ with periodic boundary conditions. They interact via the Lennard-Jones potential

$$V(r_{ij}) = 4\epsilon \left[ \left( \frac{\sigma}{r_{ij}} \right)^{12} - \left( \frac{\sigma}{r_{ij}} \right)^{6} \right], \tag{14}$$

where $r_{ij}$ is the distance between particles $i$ and $j$, the energy scales with $\epsilon$, and $\sigma$ defines the characteristic length scale. For the presented results, $\epsilon = 1$ and $\sigma = 1$. In order to decrease the computational effort, a domain decomposition is used and the interaction range is limited to a cut-off radius $r_c$:

$$V^*(r_{ij}) = \begin{cases} V(r_{ij}) - V(r_c) & r_{ij} < r_c = 2.5\sigma \\ 0 & \text{else} \end{cases}. \tag{15}$$

Since the particle momenta are independent of position, we can integrate their corresponding degrees of freedom explicitly. This contributes a constant (temperature-dependent) factor to the partition function, and canonical expectation values are thus unaffected. For a recent discussion, see [24]. Here, we only consider potential energy ($E$).

## 3.2 Metropolis Simulations

The first Markov chain Monte Carlo (MCMC) technique we employ is the METROPOLIS algorithm [25]. System configurations are generated according to the Boltzmann distribution, the corresponding configuration weight is $e^{-\beta E}$ and the canonical partition function can be written as

$$Z_{\text{NVT}} = \int dE \, \Omega(E) \, e^{-\beta E}, \tag{16}$$

where $\Omega(E)$ is the density of states and $\beta = 1/k_B T$ is the inverse temperature, as usual. A common obstacle for MCMC simulations is the critical slowing down in the vicinity of phase transitions. In case of first-order phase transitions, the probabilities of states between metastable phases are heavily suppressed and, especially when sampling with $e^{-\beta E}$, the system tends to stay a very long time in either phase.

## 3.3 Parallel Multicanonical Simulations

The multicanonical method (MUCA) [26–31] is our algorithm of choice in the fixed-density regime. Instead of sampling the energy range according to a constant temperature, as is done with METROPOLIS, it allows to cover a predefined energy range and to reweight (in a post-production step) to any desired temperature for which underlying energies were sampled. This is possible because the probability of intermediate states is artificially enhanced by replacing the Boltzmann weight with a beforehand unknown configuration weight $W(E)$, which, by construction, ensures a flat probability distribution in energy. The multicanonical partition function reads

$$Z_{\text{MUCA}} = \int dE \, \Omega(E) \, W(E) \tag{17}$$

and the weights are iteratively obtained in a recursive simulation before the production run takes place [32, 33]. In our MUCA simulations, only particle-displacement moves are performed, any of which attempts to change a particle's position – either locally within the interaction range or to a completely random position. The acceptance probability is

$$a = \min\left( 1, \frac{W(E_{\text{new}})}{W(E_{\text{old}})} \right), \tag{18}$$

where the system's energy will be updated from $E_{\text{old}} \rightarrow E_{\text{new}}$ if the proposed configuration change is accepted. Our multicanonical simulations are performed in parallel on at least 64

threads. Following the formalism presented in [34], the weight iteration is spread across multiple threads, wherein each one creates a separate histogram of occupied energies. After full sweeps, the histograms are merged and transferred to the host, where the new weights are generated and distributed back to all threads for the next iteration step. It was also shown that the parallelisation of the production run (yielding data from independent Markov chains) provides accurate results when compared to a single one with an equal amount of total measurements [15, 34].

### 3.4 Switching to the Grand Canonical Ensemble

In this work, we also use an adaptation of MUCA to the grand canonical ensemble (MUGC). Flat histogram methods of this and similar kind are commonly found in the literature [19, 35–37]. The biggest advantage of the approach outlined here is that, by design, the very same parallelised code base is used in canonical and grand canonical versions. The only small differences are the weight variables and acceptance tests [38]. Where MUCA uniformly samples a desired energy range, MUGC does the equivalent for a density range. To that end, the grand canonical partition function can be written as

$$Z_{\mu\mathrm{VT}} = \sum_{N=0}^{\infty} \int \mathrm{d}E_N \, \Omega(E_N) \, e^{-\beta E_N} \, e^{\beta\mu N} \,, \tag{19}$$

where $\mu$ is the chemical potential and the dependence of energy on the amount of particles in the system is emphasised. Analogously to the modification of $Z_{\mathrm{NVT}}$ to obtain $Z_{\mathrm{MUCA}}$, the contribution of the particle number to the configuration weight $e^{\beta\mu N}$ is replaced by artificial weights $W(N)$ that shall yield a flat distribution:

$$Z_{\mathrm{MUGC}} = \sum_{N=0}^{\infty} \int \mathrm{d}E_N \, \Omega(E_N) \, e^{-\beta E_N} \, W(N). \tag{20}$$

The probability distribution is flat with respect to $N$ this time, and we may reweight to any chemical potential (although only at the one simulated temperature).

Due to the variable particle number, grand canonical simulations are required to feature insertion and deletion moves, while displacements are only included to increase the algorithm's efficiency. For such asymmetric Monte Carlo updates, the suggestion probability is dependent on the particular move, which also leads to different acceptance criteria. To be more precise, a particle insertion ($N \to N+1$) is suggested at some random coordinate with probability $s_+ = 1/V$ and is accepted with

$$a_+ = \min\left(1, \frac{V}{N+1} \frac{W(N+1)}{W(N)} e^{-\beta\Delta E}\right), \tag{21}$$

where $\Delta E = E_{\mathrm{new}} - E_{\mathrm{old}}$. The inverse, a particle deletion, is suggested with $s_- = 1/N$; any random particle of the currently present ones is attempted to be deleted and the move gets accepted with

$$a_- = \min\left(1, \frac{N}{V} \frac{W(N-1)}{W(N)} e^{-\beta\Delta E}\right). \tag{22}$$

For particle displacements, the acceptance criterion of the METROPOLIS algorithm is used since the particle number remains unchanged. The MUGC method does not only enable us to measure the pure-phase densities $\rho_l$ and $\rho_g$, but one can also reweight the grand canonical time series. For canonical estimators, only those entries $O_i$ of the time series that match the particle number of interest are considered: $\langle O \rangle_{\mathrm{NVT}} \approx \overline{O}_{\mathrm{NVT}} = \sum_i O_i \, \delta_{N_i N} / \sum_i \delta_{N_i N}$. Such estimates

can be calculated for any $N$ that was included in the predefined density range and at no point of this procedure is an explicit knowledge of the chemical potential required.

We want to mention one more subtlety about the insertion and deletion moves: As a consequence of the changing total particle number, the average probability of selecting a particle for deletion somewhere in the memory container is not uniformly distributed across memory addresses (or indices). Subsequently, an unwanted correlation may be introduced if the insertion move systematically adds newly created particles to the end of the memory container. In our tests, this led to a small systematic shift (towards smaller droplets) of the MUGC results [39]. We avoid the relation between particle age and memory index by inserting particles at random positions (not only in the simulation volume but also in memory). To ensure a correct implementation, we have carefully checked that the reweighting results from MUGC match those from a generic, long METROPOLIS simulation.

### 3.5 Simulation Procedure

The first task at hand is to determine all required constants in the grand canonical reference ensemble for the two-dimensional Lennard-Jones gas. To that end, we start with a MUGC simulation at fixed $T = 0.4$ (and in very close vicinity at $T \pm 0.002$) sufficiently below the critical temperature $T_{\text{crit}} \approx 0.46$ [40]. For this choice of temperature, we need to cover a density range of $0 \leq \rho \leq 0.8$. Thereby, we make sure that both the liquid and the gas phase can be sampled. Otherwise, our criterion of phase coexistence cannot be fulfilled; we reweight the measured flat MUGC histogram $H(N)$ to the equal-height chemical potential, so that both pure phases are equally likely. After normalisation, this yields the grand canonical probability distribution $P(N) \propto P(\rho)$, as illustrated in Fig. 4.

Our given upper density threshold is, of course, highly dependent on temperature as it stems from the bulk-liquid peak position. Choosing the threshold too large will diminish acceptance rates of insertion moves: Packing ratios and geometric constraints become relevant once the solid phase is approached for larger densities. Too low an upper density threshold prevents sampling the full liquid peak and renders the equal-height criterion inapplicable. Similarly, selecting the simulation temperature is rather delicate. Merely lowering the temperature to $T < 0.39$ makes it impossible to distinguish whether the high-density peak represents a liquid or solid phase because the respective suppression is overlapped by the two peaks. On the other hand, raising the temperature quickly diminishes the suppression between the gas and the liquid peak and, eventually, the equal-height criterion cannot be fulfilled.

After the grand canonical reference quantities are thus found (Sec. 4.1), we return to the canonical ensemble. In all further simulations, time series are recorded for the energy $E$, the current particle number $N$ (and thereby $\rho$) as well as the number of particles contained in the droplet $N_{\text{D}}$ (where particles are counted to belong to a cluster if they are located within $2\sigma$ of another cluster particle). For the former two, we also store histograms that are updated after every attempted update. Error estimates are made using Jackknife and binning methods [41], where we directly treat the individual time series or histograms generated by the (parallel) threads as the underlying bins. Using the constants from the first part, Eqs. (4) and (6) allow us to directly map $N$ and $N_{\text{D}}$ onto the respective finite-size corrected observables: $\Delta = \Delta(\rho, V)$ and $\lambda = \lambda(\rho, V, N_{\text{D}})$.

Focusing on the droplet formation-dissolution transition, we start with the regime of fixed temperature (Sec. 4.2.1) and use the control parameter $\Delta$. Even though we are now interested in the canonical ensemble, we perform MUGC simulations, again at $T = 0.4$ – but this time on a smaller density range that covers the vicinity of the phase transition. Thereby, we end up with a (multi) grand canonical time series that can be reweighted to any sampled $N$, yielding canonical expectation values. Alternatively, this could be achieved by running various independent METROPOLIS simulations for each particle number (we only used METROPOLIS to

confirm that our canonical expectation values match across different methods). The narrowed down density range corresponds to a rescaled density of $0 < \Delta < 1.5$. By limiting the particle number more strictly, the new range allows us to reach linear system sizes up to $L = 640$ (for the full density range, only systems up to $L = 70$ were realistic).

In the fixed-density regime (Sec. 4.2.2), temperature serves as the control parameter. Hence we run MUCA simulations at increasing particle numbers and adjust the volume to match the desired density. The resulting time series can be reweighted to wanted temperatures, providing us again with canonical expectation values. In preceding works [23, 42], the density was set to $\rho = 0.01$, so that a "sufficiently dilute" gas was ensured. But due to the present grand canonical context, *choosing* the simulation density to be (close to) the gas density $\rho = 0.027857 \approx \rho_g(T = 0.4)$ provides us with an a priori estimate of the infinite-size transition temperature – namely $T = 0.4$ – which was set as a parameter in the previous steps.

Lastly, we want to briefly sketch the computational effort involved. We performed our simulations on a cluster of INTEL XEON E5-2640 V4 CPUs (2.4GHz). For METROPOLIS simulations, we used a single thread and started from pre-constructed states. Choosing $L = 320$ as a reference, we set $\sim 4 \times 10^9$ thermalisation updates and $\sim 2 \times 10^{10}$ measurement updates. This typically took $\sim 3$ days. For the corresponding parallel MUGC simulation ($L = 320$), we used 128 threads. Here, the adaptive weight iteration (including thermalisation) required $\sim 6$ hours. The consecutive production run took $\sim 3$ days for $\sim 9 \times 10^{10}$ updates per thread. For the comparable parallel MUCA simulation ($L \approx 380$), we used 240 threads. Here, the adaptive weight iteration (including thermalisation) required $\sim 6$ hours. The following production run took $\sim 1$ day for $\sim 8 \times 10^{10}$ updates per thread. As an upper maximum, the $L = 640$ METROPOLIS simulations ran for up to 70 days. The most extensive parallel MUCA simulation took $\sim 16$ days for $N = 12288$ ($L \approx 660$) on 240 threads.

## 4 Results

### 4.1 Grand Canonical Reference Quantities

Using the parameters outlined in the previous section, equally high pure-phase peaks of the grand canonical probability distribution are possible. Since density is discretised through the particle number, it is convenient to stay in the representation via $N$. The probability minimum at $N_s = \rho_s V$ separates the two peaks, each of which can be in leading order approximated as a Gaussian [14, 20]. The according particle numbers are

$$N_g = \langle N \rangle_g = \sum_{N=0}^{N_s} N P(N) \bigg/ \sum_{N=0}^{N_s} P(N) \tag{23}$$

and

$$N_l = \langle N \rangle_l = \sum_{N=N_s}^{N_{\max}} N P(N) \bigg/ \sum_{N=N_s}^{N_{\max}} P(N), \tag{24}$$

where $N_{\max}$ is the largest particle number that was allowed in the simulation. At fixed temperature, the peak positions stay *almost* constant for changing system sizes. That is, for the rescaling in leading order that they contribute to, the finite-size corrections to those grand canonical observables themselves are not dominant enough to have an impact [Figs. 4 and 5 a, b)]. As a consequence, finite-size corrections to the bulk densities are negligible.

We further obtain the (gas) peak width – corresponding to the variance of the respective expectation value – which is a measure for the reduced isothermal compressibility:

$$\hat{\kappa} = \frac{\beta}{V} \left( \langle N^2 \rangle_g - \langle N \rangle_g^2 \right), \tag{25}$$

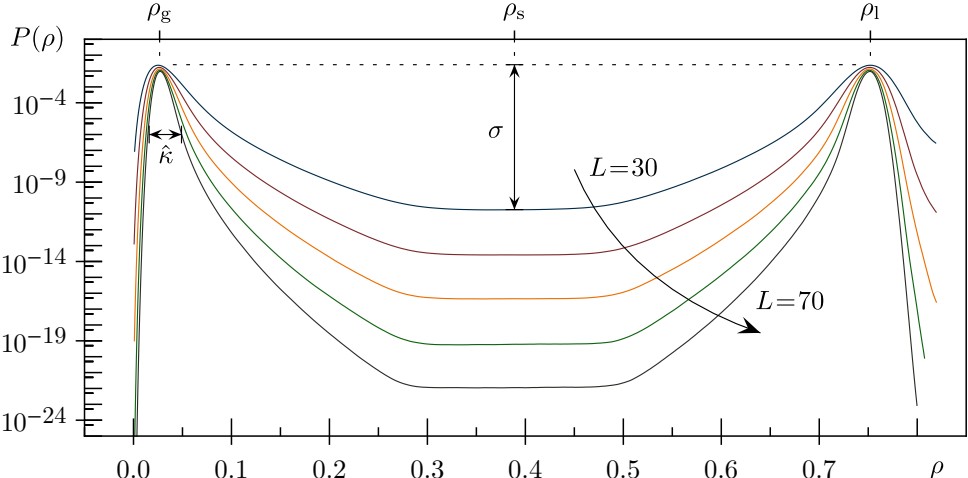

Figure 4: Grand canonical probability distribution of density at increasing linear system size $L = 30, 40, ..., 70$ and $T = 0.4$ at equal-height chemical potential. Peak positions in terms of density stay *almost* the same, while the systematic decrease in peak width – and seemingly in $\hat{\kappa}$ – is accounted by the relation between density and particle number; the amount of possible $N$-values for a given density interval increases with system size so that the compressibility stays constant after all. The depth of the probability suppression at $\rho_s$ is a measure for the linear interface tension $\sigma$. It stems from the single liquid strip spanning across the system through periodic boundary conditions, as depicted in Fig. 2.

with $\langle N^2 \rangle_g$ calculated analogously to Eq. (23). Only the compressibility that belongs to the left-hand peak is used in the subsequent rescaling. Note that $\hat{\kappa}$ is primarily a response function. The *real* isothermal compressibility in its physical sense can be related via $\kappa = -V^{-1}(\partial V/\partial p)_T = \hat{\kappa}/\rho^2$, where $\hat{\kappa}$ and $\rho$ belong to either one of the two pure phases [39]. Again, finite-size corrections to $\hat{\kappa}$ are negligible [Fig. 5 c)].

The final information we extract from the probability distribution is the depth of the suppression, from which the (normalised) interface tension can be calculated [43]:

$$\sigma = \frac{1}{2\beta L} \ln\left[\frac{P(\rho_g)}{P(\rho_s)}\right]. \tag{26}$$

As the suppression increases with system size, systematic finite-size behaviour of the form $\sigma = \sigma_\infty + aL^{-1}$ is observed [Fig. 5 d)]. The surface free energy per unit volume of the ideally shaped droplet is related as $\tau = 2\sqrt{\pi}\sigma$. However, the additive corrections to $\sigma$ will influence only higher-order corrections of $\Delta$ when plugged into Eq. (6). Hence, we stick to convention [14] and use the infinite-size estimate $\tau = 2\sqrt{\pi}\sigma_\infty$. For a collection of all discussed grand canonical reference quantities, see Table 1.

At this point, we may apply the rescaling and calculate the numerical values of the amplitudes of the leading-order scaling in both regimes. From Eq. (9), we directly calculate $a_\rho$ at fixed temperature. For fixed densities on the other hand, we have to estimate $a_T$ from Eq. (13) via $f'(\rho, T_g)$ – which requires us to assess the temperature-dependence of the reference quantities. Again using the relation that $\rho_g(T_g) = \rho$, the temperature derivative of $f(\rho, T)$ simplifies significantly and it is easy to check that

$$f'(\rho, T_g) = -\rho_g'\left(\frac{\sqrt{\rho_1 - \rho_g}}{2\hat{\kappa}\tau}\right)^{2/3}. \tag{27}$$

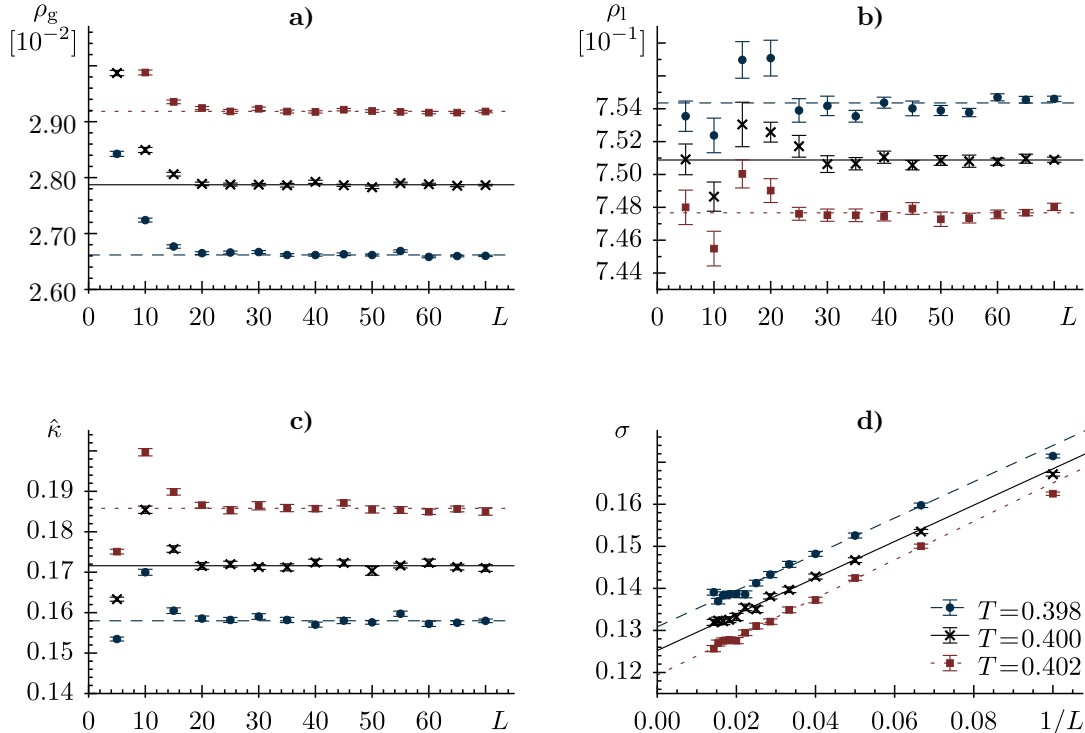

Figure 5: Grand canonical observables at fixed $T = 0.398$ (blue circles, dashed lines), $T = 0.400$ (black crosses, solid lines) and $T = 0.402$ (red squares, dotted lines): **a)** Bulk gas density $\rho_g$, corresponding to the left-hand peak of the grand canonical probability distribution. **b)** Bulk liquid density $\rho_l$, as obtained form the right-hand peak. **c)** Reduced isothermal compressibility $\hat{\kappa}$, measured as the width of the gas peak. **d)** Interface tension $\sigma$, plotted and linearly fitted as a function of inverse system size. The intersection with $1/L = 0$ is used for rescaling.

Table 1: Results for the grand canonical reference quantities. Systems of linear size $L \geq 20$ were included in the least-square fits (Fig. 5).

| $T$ | $\rho_g$ | $\rho_l$ | $\hat{\kappa}$ | $\sigma_\infty$ |
|---|---|---|---|---|
| 0.398 | 0.0266170(55) | 0.754354(85) | 0.15799(17) | 0.13087(53) |
| 0.400 | 0.0278723(62) | 0.750881(87) | 0.17160(22) | 0.12523(54) |
| 0.402 | 0.0291850(69) | 0.747667(90) | 0.18580(23) | 0.11937(50) |

By evaluating the grand canonical observables slightly above and below our reference temperature of $T = 0.4$, we may approximate $\rho_g' = \partial\rho_g/\partial T \approx \Delta\rho_g/\Delta T$, where $\Delta\rho_g = \rho_g(T = 0.402) - \rho_g(T = 0.398)$ and $\Delta T = 0.004$. The amplitudes for the leading-order scaling of density and temperature are then

$$a_\rho = \left( \frac{2\hat{\kappa}\tau\Delta_c}{\sqrt{\rho_l - \rho_g}} \right)^{2/3} \approx 0.300 \quad \text{and} \quad a_T = \frac{\Delta_c^{2/3}}{f'(\rho, T_g)} = -\frac{a_\rho}{\rho_g'} \approx -0.467, \quad (28)$$

respectively. From here, we also obtain a prediction for the amplitude of the scaling of the

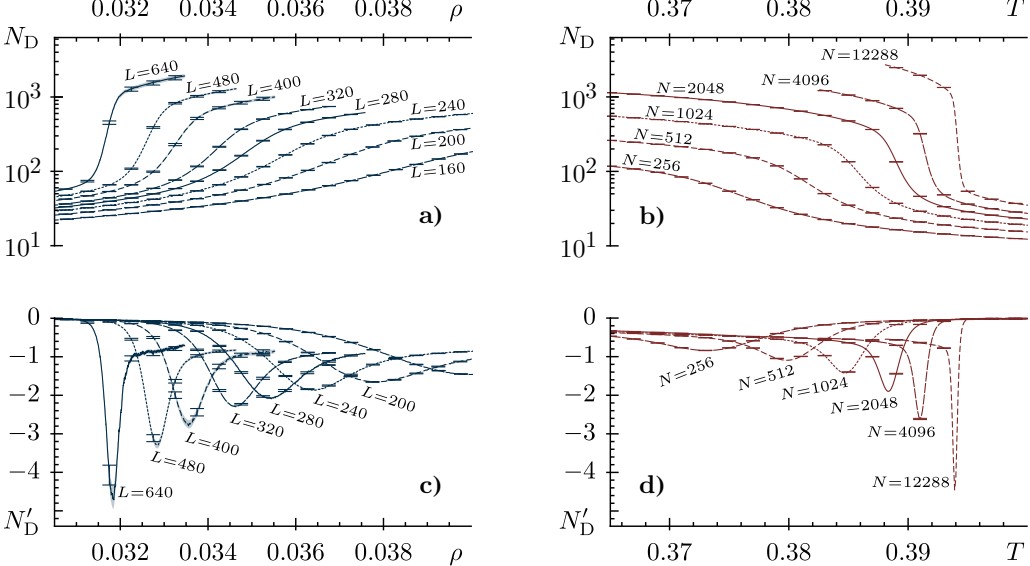

Figure 6: Scaling behaviour of the droplet formation-dissolution transition at fixed temperature $T = 0.4$ (left) and fixed density $\rho = 0.027857$ (right): **a, b)** Droplet size $N_\mathrm{D}$ as function of density and temperature, respectively. As the system size is increased, the characteristic first-order discontinuity emerges analogously in both regimes and $N_\mathrm{D}$ is of similar magnitude for comparable system sizes. **c, d)** As described in the text, $N_\mathrm{D}'$ is a proxy of the slope of $N_\mathrm{D}$ and its peak positions indicate the finite-size transition points.

droplet particle number at the transition:

$$a_{N_\mathrm{D}} = a_\rho \lambda_\mathrm{c} \left( \frac{\rho_\mathrm{l}}{\rho_\mathrm{l} - \rho_\mathrm{g}} \right) \approx 0.207 \,. \tag{29}$$

## 4.2 Finite-Size Scaling of the Droplet Formation-Dissolution Transition

We now turn to discuss the droplet formation-dissolution transition. As shown in Fig. 6 a) and b), the first-order transition behaviour is strikingly similar in both regimes. Using the number of particles in the largest droplet as the observable, the transition develops analogously at fixed temperature and fixed density.

In order to locate the transition in a consistent way across schemes, we measure the peak position of a quantity that is motivated by specific heat. More precisely, we introduce

$$N_\mathrm{D}' = \frac{1}{T^2 V} \big( \langle N_\mathrm{D} E \rangle - \langle N_\mathrm{D} \rangle \langle E \rangle \big) \tag{30}$$

to describe the fluctuations of $N_\mathrm{D}$, where expectation values are estimated (as usual) by mean values. At fixed density, $N_\mathrm{D}'$ is explicitly related to the temperature derivative: $N_\mathrm{D}' = (1/V) \partial \langle N_\mathrm{D} \rangle / \partial T$. In other words, $N_\mathrm{D}'$ is just the derivative of the order parameter with respect to the chosen control parameter in this regime. A similar relation holds at fixed temperature, although with other prefactors that lead to differently high peaks: $N_\mathrm{D}' \propto -(1/V) \partial \langle N_\mathrm{D} \rangle / \partial \rho$. The actual amplitude of the derivative depends on $\partial E / \partial \rho$, but we have verified that the peak positions of $N_\mathrm{D}'$ indeed coincide with those of the numerical derivative. In the end, we chose to use $N_\mathrm{D}'$ computed from the fluctuations as the transition criterion; this proved to be more consistent than the true numerical derivative – which is very sensitive to noisy data and requires a manual choice for the width of the (five-point) stencil.

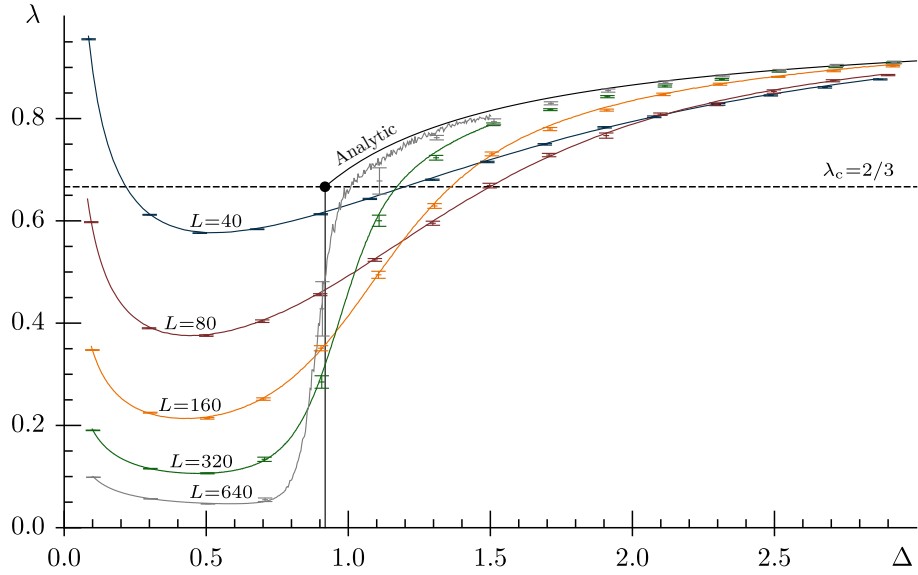

Figure 7: Droplet formation-dissolution transition at fixed $T = 0.4$, expressed in terms of the density parameter $\Delta(\rho, V)$ and the droplet fraction $\lambda(\rho, V, N_D)$ that include leading-order finite-size corrections. Individual points with error bars stem from METROPOLIS simulations and continuous lines are reweighted from MUGC. Strong finite-size effects are apparent in the gas phase but with respect to $\Delta_c$, the data convincingly approaches the analytic prediction. The intersection with the horizontal dashed line ($\lambda_c = 2/3$) is subsequently used to further investigate higher-order corrections.

#### 4.2.1 Fixing Temperature: The Oversaturated Gas

In the fixed-temperature regime [corresponding to Fig. 6 a, c)], we have a canonical ensemble in mind: We keep the temperature constant at $T = 0.4$ while increasing the particle number, ever more exceeding the bulk gas density $\rho > \rho_g$. Small amounts of particle excess ($\Delta < \Delta_c$) seemingly vanish into the oversaturation of the vapour; the gas density is only locally increased through fluctuations, but no droplet is formed. However, beyond the critical excess ($\Delta \geq \Delta_c$), free energy is no longer minimised by fluctuations alone and we observe the mixed droplet-gas phase with the majority of excess going into the droplet ($\lambda \geq 2/3$).

Contemplating Fig. 7, we can confirm that the analytic prediction is approached by the measurements as system sizes grow. This includes the curvature and the transition point; both, the threshold amount of droplet excess and the transition density move towards predicted values. In this rescaled representation, the first-order nature of the transition is visible most clearly. Evidently, our largest system ($L = 640$) suffers heavily from hidden barriers [18,36,44] and we could only record tunnel events for around 5% of the threads running in parallel for this particular size.

For small systems, notable finite-size effects in $\lambda$ are visible (and expected) for all $\Delta$. Since $\lambda$ is essentially a measure of particles within the droplet – which is at least one, even in the gas phase – small systems are prone to systematic overestimation of the droplet excess. Interestingly, these finite-size effects of $\lambda$ in the gas phase are less pronounced in three dimensions, see [15] for a comparison of the according plots for the lattice gas in two and three dimensions. We believe that this dimension-dependent behaviour can be explained by the probability suppression of particles forming intermediate clusters – which is weaker when the system is two-

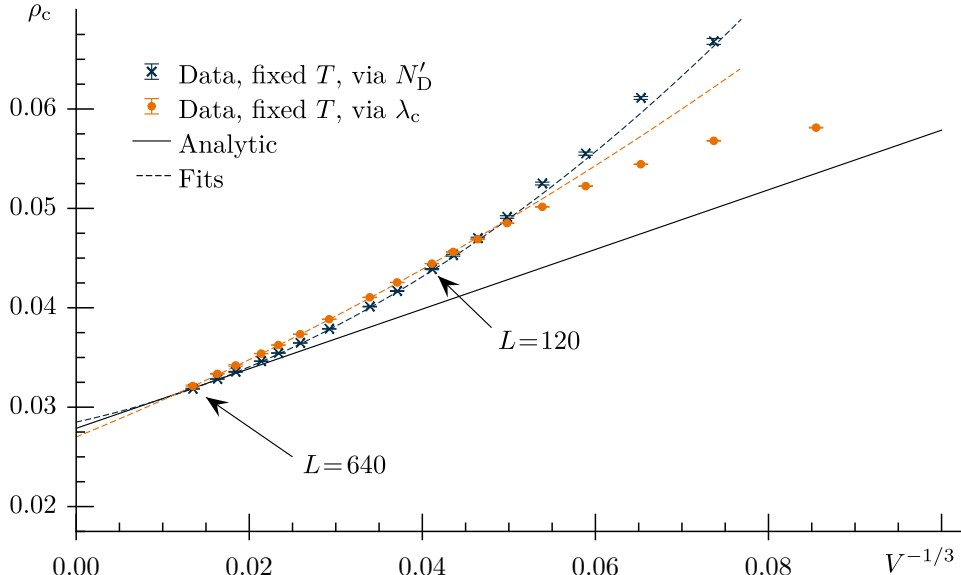

Figure 8: Finite-size scaling of the transition density $\rho_c(V)$ at $T = 0.4$. Two different criteria were employed to pinpoint the transition: For the blue crosses, the peak position of $N'_D$ was used, while the yellow circles stem from the density at which the droplet size crosses the threshold of $\lambda = \lambda_c$. The slope and offset of the analytic prediction ($\rho_c(V) = \rho_g + a_\rho V^{-1/3}$) were determined from grand canonical observables. The arrows indicate the range of data points used for the two fits ($\rho_c(V) = \tilde{\rho}_g + \tilde{a}_\rho V^{-1/3} + \tilde{b}_\rho V^{-2/3}$).

instead of three-dimensional. With respect to the transition point $\Delta_c$, we observe only weak deviations from the prediction in two dimensions.

In order to investigate these corrections to the transition density, we utilise two different approaches to specify the transition point. On the one hand, we use the intersection of $\lambda$ with $\lambda_c = 2/3$ to locate the native transition density $\rho_c(V) = \rho(\lambda_c)$, as was done in [45]. This criterion is easy to implement and since we have data for every particle number by means of MUGC, the transition density can be found with high precision. On the other hand, we take reference in the fixed-density regime, where the peak position of an observable's temperature-derivative is commonly used as an indication for the transition. Here, we are interested in the density value at which $N'_D$ is extremal (as outlined in the previous subsection). When comparing again with Fig. 7, it seems that the latter criterion – corresponding to the change in slope – is more resistant towards the systematic overestimation of droplet size.

This conjecture is confirmed in Fig. 8, which shows the scaling behaviour of $\rho_c(V)$ for both approaches. For small systems ($L < 100$), the transition point defined by $N'_D$ consistently yields higher transition densities than the $\lambda_c$-criterion. Moreover, we observe a crossover of the data points stemming from the two different approaches: beyond $L = 100$, the estimates from $N'_D$ are lower than those from $\lambda_c$ and, ultimately, approach the analytic leading order. When we only use the $N'_D$ data points of the largest three systems, a first-order fit of the form $\rho_c(V) = \rho_g + \tilde{a}_\rho V^{-1/3}$ is possible, where the tilde indicates fit parameters of the least-square fit. Here, $\tilde{a}_\rho$ is the only free parameter and $\rho_g = 0.02787$ is fixed to the grand canonical reference value. This ansatz yields $\tilde{a}_\rho = 0.2962(5)$ at $\chi^2 = 5.6$ (per degree of freedom), which is in decent agreement with the value of $a_\rho \approx 0.300$ predicted in Eq. (28).

In order to describe the behaviour for smaller systems, we empirically include the second order term ($\tilde{b}_\rho V^{-2/3}$) into the fit ansatz. One can now either fix $\rho_g$ again, or employ a fit with

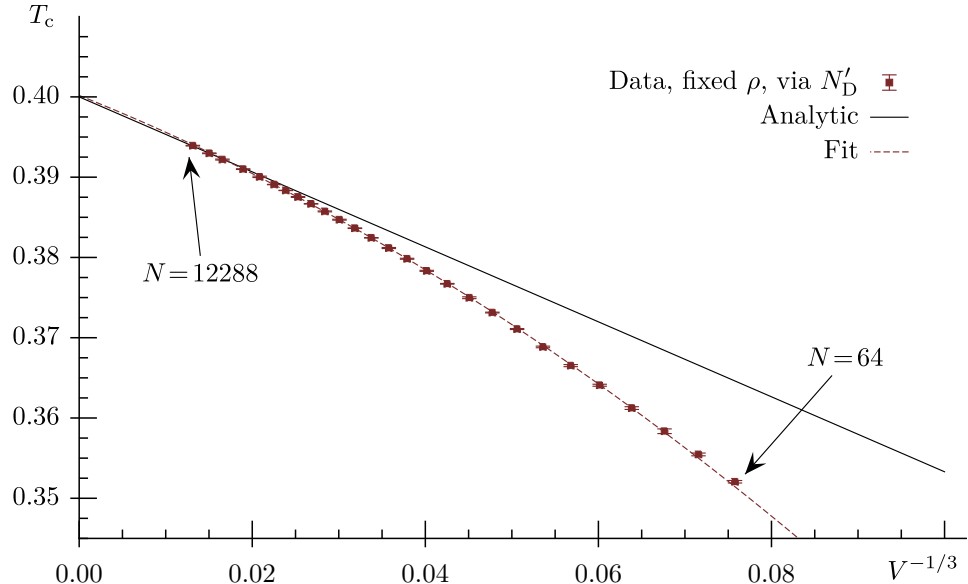

Figure 9: Scaling at fixed density $\rho = 0.027857 \approx \rho_{\mathrm{g}}(T = 0.4)$. The slope of the analytic prediction ($T_{\mathrm{c}}(V) \simeq T_{\mathrm{g}} + a_T V^{-1/3}$) is again calculated from grand canonical observables and the infinite-size transition temperature $T_{\mathrm{g}} = 0.4$ is known due to our simulation setup. All data points were included in the shown fit ($T_{\mathrm{c}}(V) = \tilde{T}_{\mathrm{g}} + \tilde{a}_T V^{-1/3} + \tilde{b}_T V^{-2/3}$).

three free parameters: $\rho_{\mathrm{c}}(V) = \tilde{\rho}_{\mathrm{g}} + \tilde{a}_\rho V^{-1/3} + \tilde{b}_\rho V^{-2/3}$. The results of the latter ansatz on the range $120 \le L \le 640$ are plotted in Fig. 8 for both data sets. Using the data from the $\lambda_{\mathrm{c}}$-criterion, the fit yields $\tilde{\rho}_{\mathrm{g}} = 0.0270(1)$, $\tilde{a}_\rho = 0.356(7)$ and $\tilde{b}_\rho = 1.6(1)$ with $\chi^2 \approx 1.5$. This fitted estimate of the infinite-size transition density $\tilde{\rho}_{\mathrm{g}}$ lies slightly below the grand canonical reference value and the amplitude $\tilde{a}_\rho$ is larger. Using the data set from the $N'_{\mathrm{D}}$-criterion, the situation changes: With $\chi^2 \approx 0.8$, the infinite-size density is overestimated as $\tilde{\rho}_{\mathrm{g}} = 0.0285(1)$, while $\tilde{a}_\rho = 0.192(9)$ is too small and $\tilde{b}_\rho = 4.3(1)$. Both fits cover respective data points of the given fit range, but in case of the $N'_{\mathrm{D}}$-data, the fit also covers small system sizes.

### 4.2.2 Fixing Density: The Undercooled Gas

Further hanging on to the canonical background, we now keep the density fixed and drive the system from gas to condensate by lowering the temperature. In particular, the simulation density was set to resemble the previously determined bulk gas density at the chosen reference temperature: $\rho = 0.027857 \approx \rho_{\mathrm{g}}(T = 0.4)$. Our MUCA simulations with set particle number (and accordingly adjusted volume) yield estimators for canonical expectation values at any temperature; the size-dependent transition temperature is then obtained from the peak-position in $N'_{\mathrm{D}}$. Analogously to the fixed-temperature regime, Fig. 9 shows our attained data points along with the analytic prediction ($T_{\mathrm{c}}(V) \simeq T_{\mathrm{g}} + a_T V^{-1/3}$), where $a_T \approx -0.467$ was again calculated from the grand canonical reference [Eq. (28)]. The plotted free fit is of similar form as before ($T_{\mathrm{c}}(V) = \tilde{T}_{\mathrm{g}} + \tilde{a}_T V^{-1/3} + \tilde{b}_T V^{-2/3}$) and fitting the complete range of data gives $\tilde{T}_{\mathrm{g}} = 0.40018(6)$ with $\tilde{a}_T = -0.430(3)$ and $\tilde{b}_T = -2.80(5)$ at good $\chi^2 \approx 1.6$.

Similarly to the fixed-temperature regime, the fit nicely covers the full range of system sizes – when the same criterion (peak-positions in $N'_{\mathrm{D}}$) is used for both. We conclude that remaining higher-order corrections must have negligible amplitudes. Having said so, an actual fit of our data to first order is only possible when restricting the data points to the largest four systems:

![SciPost]

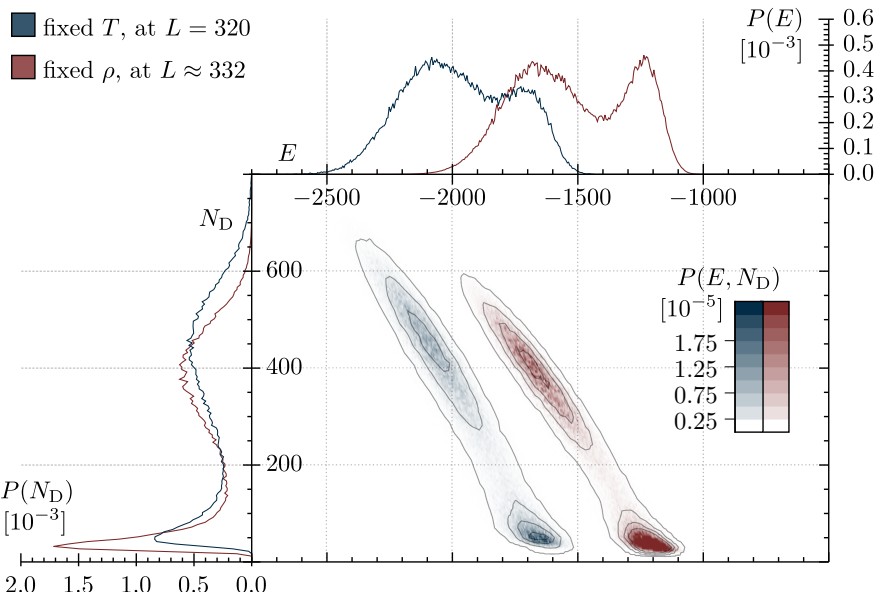

Figure 10: Comparison of probability distributions at the finite-size transition points – obtained via $N'_{\mathrm{D}}$ – for the droplet formation-dissolution transition at fixed temperature (blue) and fixed density (red). The two-dimensional distribution $P(E, N_{\mathrm{D}})$ with respect to both reaction coordinates $E$ and $N_{\mathrm{D}}$ can be projected along either axis to obtain $P(N_{\mathrm{D}})$ on the left, or $P(E)$ on the top.

Fitting $T_{\mathrm{c}}(V) = T_{\mathrm{g}} + \tilde{a}_T V^{-1/3}$ (and using systems of $3072 \leq N \leq 12288$) yields $\tilde{a}_T = -0.470(1)$ with $\chi^2 \approx 1.8$, which is in rather good agreement with the analytic prediction in Eq. (28).

Hence, we can confirm that the expected leading-order behaviour manifests at fixed density. Fits to first order are indeed possible. This was previously observed in three dimensions, in which case the leading-order behaviour manifests already for much smaller systems with only $N \leq 2048$ particles [23, 42].

### 4.2.3 Droplet Size

We now want to address the finite-size scaling of the droplet size at the finite-size transition point. Directly at the transition, the droplet phase is in coexistence with the gas phase as shown by the double-peak probability distributions in Fig. 10: The two-dimensional probability distribution $P(E, N_{\mathrm{D}})$ reveals a correlation between the reaction coordinates $E$ and $N_{\mathrm{D}}$: Energy decreases with increasing droplet size [24]. The coexistence at the droplet formation-dissolution transition thus manifests in double peaks in both distributions, $P(E)$ and $P(N_{\mathrm{D}})$.

The canonical equilibrium estimate of the droplet size $N_{\mathrm{D}}$ would be a weighted average over both gas and droplet phase. However, we seek the size of the largest cluster in the droplet phase. Hence, we measure $N_{\mathrm{D}}$ as the expectation value conditioned on the droplet phase, i.e., we only consider the right-hand (upper) peak of $P(N_{\mathrm{D}})$. One may correctly expect that there is a strong dependence on the control parameter ($\rho$ or $T$) and, for fixed $T$, whether we determine the finite-size transition density via the peak position in $N'_{\mathrm{D}}$ or as the crossing point of $\lambda(\rho) = \lambda_{\mathrm{c}}$ (cf. Fig. 8). Therefore, we obtain the expectation value of $N_{\mathrm{D}}$ as follows: When the transition point was determined via the $N'_{\mathrm{D}}$-criterion, then we first calculate $N_{\mathrm{D,eq}}$ such that $\int_0^{N_{\mathrm{D,eq}}} P(N_{\mathrm{D}}) = 1/2$ and then evaluate $N_{\mathrm{D}}$ inside the droplet phase via the *conditional* expectation value $2 \int_{N_{\mathrm{D,eq}}}^N N_{\mathrm{D}} P(N_{\mathrm{D}})$. When the transition point was determined via $\lambda_{\mathrm{c}}$, then the droplet phase dominates and the conditional expectation value practically coincides with

the equilibrium estimate.

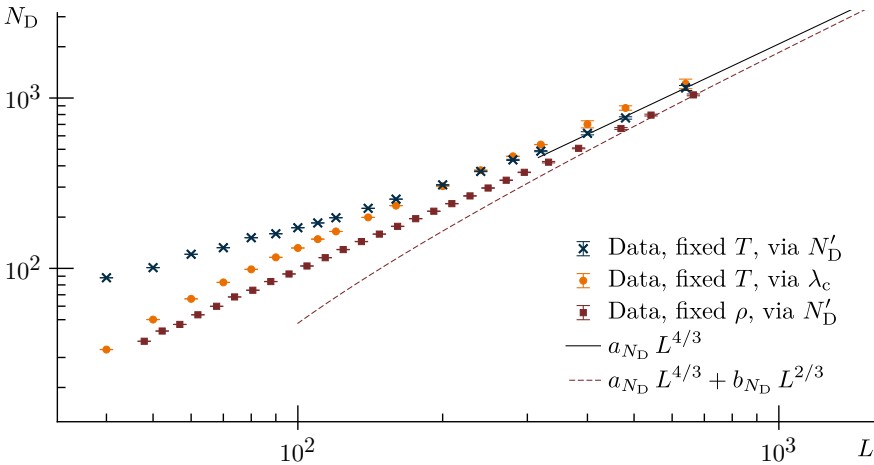

Figure 11: Scaling of the droplet size $N_D$ at the finite-size transition point for fixed temperature (blue crosses and yellow circles) and at fixed density (red squares). The analytic prediction is plotted to leading order [black, solid, Eq. (11)] along with its temperature expansion to second order [red, dashed, Eq. (32)], as is necessary to describe the fixed-density data. The amplitudes $a_{N_D}$ and $b_{N_D}$ were determined from grand canonical reference quantities.

The finite-size scaling of the droplet size is plotted in Fig. 11. Note that the leading-order behaviour predicted for fixed temperature [Eq. (11), $N_D(\rho_c(V), V) = a_{N_D} V^{2/3}$] is ultimately approached by all data sets (from the two criteria and both regimes).

At fixed temperature, finite-size effects become small once the linear system size reaches $L \geq 200$. As for the transition density in Fig. 8, we observe a crossover of the data from the two different transition criteria. Having said so, the results from both approaches converge towards the leading order prediction rather quickly and the difference is barely visible on the shown scale.

At fixed density, the droplet size is systematically smaller than at fixed temperature but shows the same $V^{2/3} = L^{4/3}$ trend. In fact, the leading-order scaling of the droplet size in Eq. (11) was (so far) only given for fixed temperature. To derive the finite-size scaling of the droplet size at fixed density, we go back to Eq. (10) and re-introduce the temperature dependence:

$$N_D(\rho, V)/\lambda V = \left(\rho - \rho_g(T)\right)\left(\frac{\rho_l(T)}{\rho_l(T) - \rho_g(T)}\right) = g(\rho, T). \tag{31}$$

When expanding $g(\rho, T)$ to second order around the infinite-size transition temperature $T_g$, most terms vanish because $\rho_g(T_g) = \rho$. We then plug in the leading-order scaling for fixed density [Eq. (13), $(T_c(V) - T_g) \simeq a_T V^{-1/3}$] – supported by Fig. 9, from which we know that the largest system sizes indeed approach this leading-order solution – and arrive at

$$N_D(T_c(V), V) \simeq \lambda_c(-\rho_g')\left(\frac{\rho_l}{\rho_l - \rho_g}\right)a_T V^{2/3} + \lambda_c\left[\rho_g'\frac{\rho_l'\rho_g - \rho_l\rho_g'}{(\rho_l - \rho_g)^2} - \rho_g''\frac{\rho_l}{\rho_l - \rho_g}\right]a_T^2 V^{1/3}$$

$$\simeq a_{N_D} V^{2/3} + \lambda_c\left[\rho_g'\frac{\rho_l'\rho_g - \rho_l\rho_g'}{(\rho_l - \rho_g)^2} - \rho_g''\frac{\rho_l}{\rho_l - \rho_g}\right]\left(\frac{a_\rho}{\rho_g'}\right)^2 V^{1/3} \tag{32}$$

$$\simeq a_{N_D} V^{2/3} + b_{N_D} V^{1/3},$$

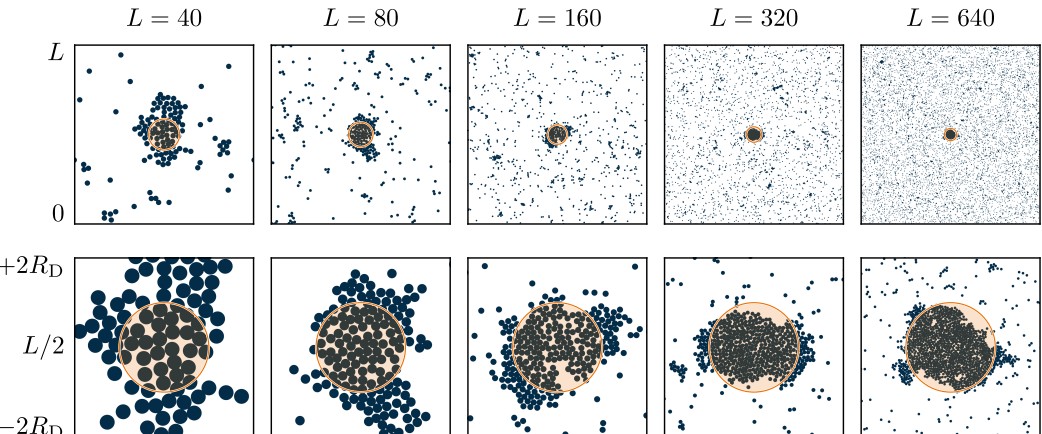

Figure 12: Example configurations at fixed temperature $T = 0.4$ for increasing system sizes. The snapshots were taken so that $\rho$ and $N_D$ attain their size-dependent transition values, obtained via $N_D'$ (i.e. matching the blue crosses in Fig. 11, not the analytic prediction). Although the droplet grows continuously with system size, its relative volume vanishes as the probability gas peak sharpens (**top row**). With increasing system size, the observed droplet volume and shape approach the analytic prediction with radius $R_D = (a_{N_D}/\pi\rho_l)^{1/2} L^{2/3}$, shown as shaded circle (**bottom row**, rescaled to $R_D$).

where we used $a_T = -a_\rho/\rho_g'$ and $a_{N_D} = a_\rho \lambda_c \rho_l/(\rho_l - \rho_g)$. Thus, the leading-order amplitude in the finite-size scaling of the droplet size at fixed density coincides with that at fixed temperature – if the infinite-size transition point $(\rho, T)$ coincides. This initially seems quite surprising but is explained by the finite-size transition points converging to the same limit with increasing system size, while the droplet continues growing. Note that, even though the droplet grows to infinity, its relative size (compared to the box size) vanishes, see Fig. 12. Using the grand canonical reference quantities – where we computed the derivatives as ratios of finite differences ($\rho' = \Delta\rho/\Delta T$ and $\rho'' = \Delta\rho'/\Delta T$) – we can evaluate $a_{N_D} \approx 0.207$ as well as $b_{N_D} \approx -2.264$. The prediction of Eq. (32) is shown as the dashed line in Fig. 11 and well describes the largest system sizes, where the actual volume occupied by the droplet does not exceed the analytic prediction.

## 5 Conclusion

We have verified the leading-order theory on equilibrium droplet formation and dissolution [10–12] for the two-dimensional Lennard-Jones gas at fixed temperature (varying density), and at fixed density (varying temperature). Specifically for fixed temperature, we showed that the analytic prediction by Biskup et al. [10] well describes the size of the largest droplet as a function of density. While this solution is rigorously proven for the lattice gas, we are not aware of prior confirmations for continuous systems. For the orthogonal case of fixed density, we adapted the theory to obtain an analytic prediction for the leading-order scaling of the transition temperature [23]. Using grand canonical reference values, we were able to quantitatively predict the amplitude of the leading-order corrections. In particular, we found a direct relation between those amplitudes of corrections on the transition density, the transition temperature, and the transition droplet-size. Surprisingly, we found that the scaling of the finite-size transition density and temperature down to very small system sizes is well

described by the leading-order term $V^{-1/3}$ plus a heuristic (quadratic) higher-order correction term $V^{-2/3}$, despite knowing that there is a multitude of higher-order correction sources, including capillary waves, the Gibbs-Thompson effect, the breakdown of the Gaussian approximation, and logarithmic corrections [11, 13, 14, 24, 46].

Most importantly, we showed that a switch between control parameters (here density and temperature) is straightforward, such that numerical approaches may fall back onto the setup most easily realised. For example, with macromolecules it is very easy to work in the canonical ensemble [24], where the orthogonal setup in the grand canonical ensemble is more involved [47, 48]. Of course, combining both approaches allows one to estimate higher-order corrections consistently, which provides a complete picture of the finite-size scaling behaviour.

In order to obtain the precise data presented in this study, we applied parallel generalised-ensemble simulations in the (multi) canonical and (multi) grand canonical ensemble. The general formulation of the method presented in Sec. 3 should allow an easy application of this powerful parallel method to other setups, in particular those involving nucleation-like transitions. In fact, it was shown that the parallelisation scales very well up to $O(10^5)$ threads and it can be implemented on both CPU and GPU clusters [49]. Examples of nucleation-like problems that benefited from this method include polymer aggregation [24] as well as formation of void-spaces in the Blume-Capel model – a model for superfluidity in $^3$He–$^4$He mixtures [50, 51] – where the generalised ensemble can be adapted to the crystal-field [52]. Parallel multicanonical simulations should also be very fruitful for the study of heterogeneous nucleation at flat and structured surfaces [53].

Our approach may thus serve as a template for the study of other nucleation-like problems. Examples include cluster formation in colloidal, polymer and protein solutions [24, 54, 55], crystallisation in colloidal suspensions [55, 56], nucleation in iron melts [57], so-called phase-change materials [58–60] and glassy solids [61], as well as domain formation in ferromagnetic materials [62] or mixtures [50–52].

Lastly, we note that apart from their physical relevance in surface science, two-dimensional systems are important model systems for the study of generic properties accompanying nucleation. We believe that our results may serve as a reference point, e.g., for the study of free-energy barriers in the presence of nucleation seeds, or to resolve the question about the "critical" initial droplet size in equilibrium droplet formation. Another advantage of two-dimensional models is the straightforward usage of transition-path methods such as the string method [44, 63, 64], as well as an easy control of geometric parameterisations. In combination with the advanced parallel generalised-ensemble methods we presented here, this may prove helpful for tackling some of the long-standing questions about nucleation.

## Acknowledgements

**Funding information**  The project was funded by Deutsche Forschungsgemeinschaft (DFG) under Grant No. JA 483/31-1. JZ received financial support from the German Ministry of Education and Research (BMBF) via the Bernstein Center for Computational Neuroscience (BCCN) Göttingen under Grant No. 01GQ1005B.

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
