# Peer review of "The Droplet Formation-Dissolution Transition in Different Ensembles: Finite-Size Scaling from Two Perspectives"

_SciPost Physics, doi:SciPost Phys. 5, 062 (2018)_

## Round 1 · Referee Report · Daniele Coslovich (Referee 1) · 2018-7-24

Strengths

1) Comprehensive finite-size scaling analysis of equilibrium droplet formation 2) Solid numerical work 3) Beautifully presented

Weaknesses

1) Generalizing the approach beyond the simple case considered here may not be straightforward

Report

The authors report extensive multicanonical Monte Carlo simulations of droplet formation in a 2d Lennard-Jones gas. They focus on the equilibrium properties of the largest droplet using either density or temperature as control parameter and perform a careful finite-size scaling analysis. The results are compared to the phenomenological theory of Biskup et al. (Ref.10), which is found to describe well the droplet scaling properties for large system sizes. The authors argue that multicanonical simulations combined with their finite-size scaling setup might prove useful in the study of more general nucleation-related problems.

The paper is extremely well written and the results are beautifully presented. The figures are remarkable for clarity and aesthetics. Scientifically, the work is solid and has the merit of discussing finite-size scaling of droplet formation within a coherent framework that treats density and temperature on similar grounds. Most previous works only focused on either one or the other control parameter. However, it is difficult for me to judge on the usefulness of the present approach beyond the specific application studied here (a simple 2d LJ gas). I only have a few remarks that the authors may want to take into account before publication.

Requested changes

1) The physical origin of the deviations from the theoretical finite-size scaling (figs. 8-11) remains unclear (higher-order corrections or other effects?). In particular, the possible artifacts due to periodic boundary conditions are not explicitly discussed. Can the authors give some further insight into this? It would also be useful to show selected samples showing the typical shape of the largest droplets for different system sizes.

2) It would be useful to provide some details on the length of the multicanonical simulations (MC steps, wall time). How long does it take to ensure an equilibrium sampling of the droplet properties at the largest system sizes?

3) At the end of p. 9, the authors mention a potential flaw in grand-canonical MC simulations, which can be fixed by inserting particles at random not only in physical space but also in the allocated area of computer memory. How bad does the simulation fail if this is not enforced? Subtle violations of detailed balance may be difficult to detect.

  • validity: top
  • significance: good
  • originality: good
  • clarity: top
  • formatting: perfect
  • grammar: perfect

Author:  Franz Paul Spitzner  on 2018-10-29  [id 334]

(in reply to Report 1 by Daniele Coslovich on 2018-07-24)

We thank the referee for the extensive and positive feedback and we would like to respond to all suggestions point-by-point, below.

The paper is extremely well written and the results are beautifully presented. The figures are remarkable for clarity and aesthetics. Scientifically, the work is solid and has the merit of discussing finite-size scaling of droplet formation within a coherent framework that treats density and temperature on similar grounds. Most previous works only focused on either one or the other control parameter. However, it is difficult for me to judge on the usefulness of the present approach beyond the specific application studied here (a simple 2d LJ gas).

The presented realisation of our approach is tailored to the problem at hand for which it was designed. It is, however, indeed straightforward to be applied to general nucleation-like problems with first-order phase transitions as we have shown in previous works and stated in the Conclusion of our manuscript as: "Examples of nuleation-like problems that benefited from this method include polymer aggregation [24] as well as formation of void-spaces in the Blume-Capel model – a model for superfluidity in 3 He– 4 He mixtures [49, 50] – where the generalised ensemble can be adapted to the crystal-field [51]."

1) The physical origin of the deviations from the theoretical finite-size scaling (figs. 8-11) remains unclear (higher-order corrections or other effects?). In particular, the possible artifacts due to periodic boundary conditions are not explicitly discussed. Can the authors give some further insight into this? It would also be useful to show selected samples showing the typical shape of the largest droplets for different system sizes.

We followed the suggestion and added Fig. 12 to the manuscript that depicts system configurations for increasing system sizes and added in the text accordingly: "This initially seems quite surprising but is explained by the finite-size transition points converging to the same limit with increasing system size, while the droplet continues growing. Note that, while the droplet grows to infinity, its relative size (compared to the box size) vanishes, see Fig. 12. Using the grand canonical reference quantities, we can evaluate aND ~= 0.207 as well as bND ~= −2.264. The prediction of Eq. (32) is shown as the dashed line in Fig. 11 and well describes the largest system sizes, where the actual volume occupied by the droplet does not exceed the analytic prediction."

As to the physical origin of higher-order corrections, we briefly commented on the considerable amount of well-known sources in the conclusion, which we now extended by respective references (Ref. 46 is new): "[...], despite knowing that there are a multitude of higher-order correction sources, including capillary waves, the Gibbs-Thompson effect, the breakdown of the Gaussian approximation, and logarithmic corrections [11,13,14,24,46]."

2) It would be useful to provide some details on the length of the multicanonical simulations (MC steps, wall time). How long does it take to ensure an equilibrium sampling of the droplet properties at the largest system sizes?

We have estimated that the total computing time of the presented data (excluding test runs) amounts to fifty core years. We further added a short note on the computational effort to the manuscript: "Lastly, we want to briefly sketch the computational effort involved. We performed our simulations on a cluster of Intel Xeon E5-2640 v4 CPUs (2.4GHz). For Metropolis simulations, we used a single core and started from pre-constructed states. Choosing L=320 as a reference, we set ~4x10^9 thermalisation updates and ~2x10^10 measurement updates. This typically took ~3 days. For the corresponding parallel Mugc simulation (L=320), we used 128 threads. Here, the adaptive weight iteration (including thermalisation) required ~6 hours. The consecutive production run took ~3 days for ~9x10^10 updates per thread. For the comparable parallel Muca simulation (L~=380), we used 240 threads. Here, the adaptive weight iteration (including thermalisation) required ~6 hours. The following production run took ~1 day for ~8x10^10 updates per thread. As an upper maximum, the L=640 Metropolis simulations ran for up to 70 days. The most extensive parallel Muca simulation took ~16 days for N=12288 (L~=660) on 240 threads."

3) At the end of p. 9, the authors mention a potential flaw in grand-canonical MC simulations, which can be fixed by inserting particles at random not only in physical space but also in the allocated area of computer memory. How bad does the simulation fail if this is not enforced? Subtle violations of detailed balance may be difficult to detect.

We noted this issue when comparing multi-grand-canonical data with Metropolis in Fig.~7. A systematic shift (towards smaller droplets) of the multi-grand-canonical MC results occurred when storing new particles always at the end of the array, while deleting random particles from the array. We now added a brief remark on this shift in the text. Currently, we believe the issue is connected to how the fluctuating particle number influences the probability of array-indices for deletions (Ref.~[39], Fig. 4.1), but we leave this to future work.

---

## Round 1 · Referee Report · Anonymous (Referee 2) · 2018-8-30

Strengths

  1. In-depth analysis of evaporation-condensation transition for the 2d Lennard-Jones system.
  2. Quality of simulation data.

Weaknesses

None

Report

The manuscript “The Droplet Formation-Dissolution Transition in Different Ensemble: Finite-Size Scaling from Two Perspectives“ by Spitzner et al investigates the system-size dependent evaporation-condensation transition of a single droplet from a homogeneous phase in the two dimensional Lennard-Jones system. The authors take great care to analyze finite-size dependencies of a number of quantities to determine the exact location of the transition points. To this extent they employ state-of-the art parallelized MUCA Monte Carlo simulations in the canonical and grandcanonical ensemble and approach the problem by changing density while keeping temperature fixed and vice versa. Results are compared to theoretical predictions.

The paper is well-written and a pleasure to read and I fully recommend publication of this manuscript in its present form.

Requested changes

If they like, the authors may consider the following minor remarks in their final version:

  1. I’d be interested in the computational effort involved in this endeavor. Maybe the authors can comment on this point.

  2. The finite-size dependence of the isothermal compressibility (Fig.5c) looks rather peculiar for the smallest systems under investigation considering the small statistical errors.

  3. Typo: page 3: “choosing any” instead of “anyone”.

  • validity: top
  • significance: good
  • originality: high
  • clarity: high
  • formatting: perfect
  • grammar: excellent

Author:  Franz Paul Spitzner  on 2018-10-29  [id 333]

(in reply to Report 2 on 2018-08-30)

We thank the referee for the attentive inspection of our submission and the positive feedback that was provided. Below, we address suggestions point-by-point.

1. I’d be interested in the computational effort involved in this endeavor. Maybe the authors can comment on this point.

We have estimated that the total computing time of the presented data (excluding test runs) amounts to fifty core years. We further added a short note on the computational effort to the manuscript: "Lastly, we want to briefly sketch the computational effort involved. We performed our simulations on a cluster of Intel Xeon E5-2640 v4 CPUs (2.4GHz). For Metropolis simulations, we used a single core and started from pre-constructed states. Choosing L=320 as a reference, we set ~4x10^9 thermalisation updates and ~2x10^10 measurement updates. This typically took ~3 days. For the corresponding parallel Mugc simulation (L=320), we used 128 threads. Here, the adaptive weight iteration (including thermalisation) required ~6 hours. The consecutive production run took ~3 days for ~9x10^10 updates per thread. For the comparable parallel Muca simulation (L~=380), we used 240 threads. Here, the adaptive weight iteration (including thermalisation) required ~6 hours. The following production run took ~1 day for ~8x10^10 updates per thread. As an upper maximum, the L=640 Metropolis simulations ran for up to 70 days. The most extensive parallel Muca simulation took ~16 days for N=12288 (L~=660) on 240 threads."

2. The finite-size dependence of the isothermal compressibility (Fig.5c) looks rather peculiar for the smallest systems under investigation considering the small statistical errors.

This seemingly 'random' behaviour is due to drastic finite-size effects. We estimate the reference quantities under the assumption of Gaussian shaped probability peaks, in particular, kappa as the peak width. The errors for all these observables stem from jackknifing the different realisations (here threads) so that the errors are rather small if all threads yield consistent results. However, for L < 20 the limited resolution on the density axis prevents the assumed Gaussian. For L = 5, the gas density corresponds to less than a single particle in the system. Hence, the probability distribution shows a sharp edge when jumping from 0 to 1 to 2 particles. Clearly, the statistical error bars do not cover this physical limitation.

3. Typo: page 3: “choosing any” instead of “anyone”.

Corrected in the new manuscript. "For such a system, we could induce droplet formation by choosing any of the three as a control parameter."

---

## Editorial Decision

published